# JFP: Joint Future Prediction with Interactive Multi-Agent Modeling for Autonomous Driving

**Wenjie Luo**[*]
wenjieluo@waymo.com

**Cheolho Park**[*]
cheolhop@waymo.com

**Andre Cornman**
cornman@waymo.com

**Benjamin Sapp**
bensapp@waymo.com

**Dragomir Anguelov**
dragomir@waymo.com

**Abstract:** We propose *JFP*, a Joint Future Prediction model that can learn to generate accurate and consistent multi-agent future trajectories. For this task, many different methods have been proposed to capture social interactions in the encoding part of the model, however, considerably less focus has been placed on representing interactions in the decoder and output stages. As a result, the predicted trajectories are not necessarily consistent with each other, and often result in unrealistic trajectory overlaps. In contrast, we propose an end-to-end trainable model that learns directly the interaction between pairs of agents in a structured, graphical model formulation in order to generate consistent future trajectories. It sets new state-of-the-art results on Waymo Open Motion Dataset (WOMD) for the interactive setting. We also investigate a more complex multi-agent setting for both WOMD and a larger internal dataset, where our approach improves significantly on the trajectory overlap metrics while obtaining on-par or better performance on single-agent trajectory metrics.

## 1 Introduction

Motion forecasting of multiple agents is of particular importance in the autonomous vehicle (AV) domain, since safe and comfortable driving requires an understanding of the future trajectory distribution of all agents on the road including but not limited to vehicles, pedestrians and cyclists.

This interesting and high-impact motion forecasting task has resulted in many proposed models to date (*e.g.* [1, 2, 3, 4, 5, 6, 7, 8, 9, 10, 11, 12, 13, 14, 15, 16, 17, 3, 18, 19]), spurred by the release of multiple popular public benchmarks for this problem [20, 21, 22, 23]. Most of the focus to-date has been on input representations and encoding of the complex world states, including an agent's relationships with other agents and the road network. With a few exceptions [19, 24], there has been almost no focus on output representations to model the joint probability of multiple agents. This is driven at least in part by the primary metrics in most major public benchmarks, which are based on trajectory distance error to the ground-truth future trajectories, computed independently for each agent.

Letting $\mathbf{s}_i$ be the future trajectory for the $i^{th}$ agent in the scene, the *marginal distribution* of $\mathbf{s}_i$ is typically represented by $K$ future trajectories with corresponding probabilities: $\{(p(\mathbf{s}_i^j), \mathbf{s}_i^j)\}_{j=1}^K$. There are clear shortcomings for using marginal prediction for autonomous driving. As a simple illustrative example shown in Fig. 1 (a), assume two vehicle agents coming into an intersection with equal prior probability of going straight or turning left. Representing the future set of outcomes with marginal probabilities, one cannot tell that two of four possible outcomes in {Agent A, Agent B} $\times$ {going straight, turning left} is infeasible. In practice, $K$ is often set to be 6 [22, 20], and there could be even more unrealistic predictions when considering more than two agents.

Instead, one could consider modeling the *joint distribution* of all $A$ agents in a scene as $p(\mathbf{s}_1, \mathbf{s}_2, \ldots, \mathbf{s}_A)$. While we keep the same number of states $K$ for each agent, the total size of

---

[*]Equal contribution

6th Conference on Robot Learning (CoRL 2022), Auckland, New Zealand.

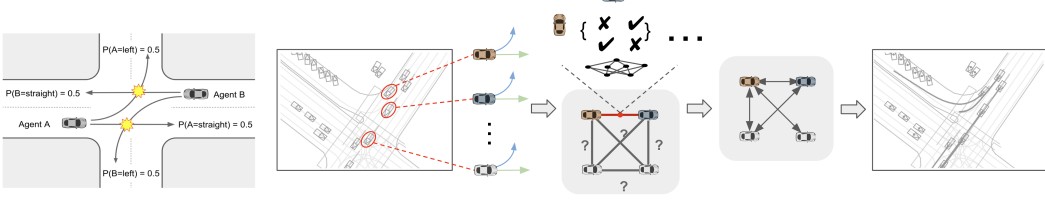

| a: Simple Example | b: Model Illustration |

Figure 1: (a): A simple two-agent example showcasing the importance of using a joint model that can output consistent future trajectories. (b): We propose to formulate the problem using pairwise graphical model, where the model first generates trajectories for all agents as unary (Sec. 3.2) and constructs the underlying interaction graph, pairwise potential is computed for the selected pairs followed by message passing to get final joint trajectories (Sec. 3.3).

the output space grows exponentially w.r.t. number of agents, $i.e.$ $K^A$. This is computationally intractable, given that common scenarios can have dozens or hundreds of agents. Towards this full joint modeling, the research community has recently explored some limited versions: the Waymo Open Motion Dataset [20] provides a joint prediction track that focuses only on the given pair of interactive agent $p(\mathbf{s}_1, \mathbf{s}_2)$; The INTERACTION dataset [23] has a conditional prediction track focusing on $p(\mathbf{s}_i|\mathbf{s}_j)$. Practitioners have addressed these tasks by directly modeling a pairwise output space $p(\mathbf{s}_1, \mathbf{s}_2)$ or conditional space $p(\mathbf{s}_i|\mathbf{s}_j)$ [24, 9]. While these methods are targeting two agents interaction cases, others have tried to approximate the full joint distribution directly by multi-agent trajectory sample rollouts [25, 8] or directly predicting a weighted set of $K$ joint predictions [19] where the joint output is tied and limited to queries.

To better model the joint probability in the multi-agent setting, we formulated it as a pairwise graphical model with a dynamic interactive graph, where nodes correspond to agents, and edges correspond to agents' pairwise relationships. Intuitively, the pairwise potential between each pair of agents can represent the consistency between their predicted trajectories. The underlying interaction graph can be built dynamically based on unary predicted trajectories. This is a more tractable solution compared with the one directly modeling everything jointly. As one special case, we also experiment with a star graph that intuitively takes an ego-centric approach and primarily focuses on modeling the interactions of the AV (Autonomous Vehicle) and other agents. Furthermore, using energy-based models allows us to easily perform conditional inference. This is important for real-world AV stacks since one would need to reason about joint futures conditioned on a specific AV trajectory.

We evaluate our approach using the public Waymo Open Motion dataset (WOMD) [20], where we set the new state-of-the-art for the interactive setting. Furthermore, we investigate the multi-agents setting where the model needs to predict consistent trajectories for all agents presented in the scene. Our approach achieves significant improvements on both WOMD and a much larger internal dataset.

## 2   Related Work

**Feature Encoding**   The input of the motion forecasting task involves time-series data such as the status of agents and traffic lights, as well as static data such as road graph elements. For encoding agents motion, LSTMs/GRU have been used in [26, 9, 8, 5, 27], VectorNet [14] adapted the polyline representation, laneGCN [2] utilized 1D convolution, [7] rasterized the agents' bounding boxes explicitly and used CNNs to encode the motion. On the other hand, [28, 3] took raw sensor data as input and implicitly learned the motion of agents with CNNs. Recently, attention layers have also been used to encode the motion of agents with promising results [19, 29, 30]. For encoding road graph data, while rasterized map images have been used in [5, 26, 28, 3, 1, 8], [2, 31] shows that graph convolution can be used to encode a map to better capture the relation between each map element. Furthermore, [15, 14, 9, 32, 33] utilized polylines to represent the road graph, which proved to be an effective approach. In our work, we also use polylines to represent the road graph data.

**Interaction Modeling** While DSDNet [34] proposed a fixed fully connected graph to model interaction among agents, RAIN[35] trained a RL module to output the agents' connectivity graph. [27] showed that attention can be used to learn effectively the relation between agents. SceneTransformer [19] and AutoBots [29] adapted axial attention layers [36] on agent state tensors' time and social dimensions separately, [37, 30] also adapted attention layers into their model for better overall performance. However, these models focus mostly on the interaction modeling in the encoding part to learn better agent features, the final trajectory candidates are predicted independently often using simple MLP from those agent feature embeddings. For generating joint outputs, MFP [38] uses discrete latent variables to represent the different modes of the joint future, ILVM[39] uses latent variables and GNN for interaction modeling and optimizes with ELBO, SceneTransformer [19] uses learnable queries to decode different joint futures. In contrast, we build a graphical model on top of the trajectory candidates with fully learned potentials in order to generate consistent future predictions. Different from [34, 35], we construct the interactive graph dynamically using predicted trajectories from the backbone network.

**Conditional Setting** Different previous methods have explored conditioning properties for generating joint predictions. M2I [40] focuses on agent pairs, where one of the agents is predicted as *influencer* and the other as *reactor*. Then it utilizes a conditional model to generate consistent future trajectories for the *reactor* based on the *influencer*'s marginal future predictions. MTP[41] first selects topology (homotopy) among a few predefined ones, and then predicts trajectories conditioning on that, while our method doesn't have the topology assumption/requirements. Different approaches have also investigated the setting of conditioning on AV's actions. In particular, [9] aimed to generate diverse predictions conditioned on hypothetical road lanes and interactions. [24] focused on agents' motion forecasting conditioned on future hypothetical trajectory (query) of ego-car. A similar setting has also been explored in [42], which conditions the agent predictions on the planning trajectory. In contrast, our approach is modeling the joint probability and can be converted into conditional setting easily via conditional inference on a graphical model.

## 3 Method

In this paper, we employ an encoder-decoder architecture [7, 27, 3, 19, 32] and propose a joint future prediction model for multi-agent interactive modeling (named JFP). Specifically, we take as input the past trajectories of all agents, the static road elements (i.e. road graph), as well as the dynamic road elements (i.e. traffic lights) to generate per-agent feature embeddings. These agent embeddings are fed into a decoder that learns to generate the trajectory candidates for each agent, and unary and pairwise potentials. We further apply a graphical model and run message passing (belief propagation) to combine these learned unary and pairwise potentials, and generate scene-consistent joint futures. In this section, we first explain the base model for trajectory generation and consistency learning, then introduce the graphical model formulation and the construction of the interaction graph. Finally, we cover the training procedure including losses for optimization and gradient computation. Fig. 1 (b) provides a high-level illustration of our model.

### 3.1 Input Representation

The inputs of our model consist of three parts: agent history, road graph, and traffic light status. Agent history contains each agent's 2D location, bounding box shape, yaw, velocity, agent type. We pre-process the data to be agent-centric to provide a normalized input for the neural network. This is done by centering each agent at its last timestep's location and rotating it to have zero yaw. We model the road graph using the efficient and effective polyline representation, where each polyline segment is created using the Ramer–Douglas–Peucker algorithm [43] from the raw road graph data. For each agent, we pre-select the subset of polyline segments that are closest and convert them into the corresponding agent-centric coordinate frame. The final polyline segment feature includes its 2D location, direction, length, and type. The traffic light features include type, states over time and 2D location, which have been similarly converted into the agent-centric coordinate system.

## 3.2 Backbone Network

Multiple different backbone networks have been proposed to fuse information among agents and context information based on LSTMs [44], CNNs [28, 1, 45], GNNs[31], ContextGating [32] or transformers [29, 19, 30]. Given its outstanding performance, we adopt MultiPath++ [32] as the backbone network to generate marginal predictions for each agent. The key component of Multi-Path++ is the Multiple Context Gating (MCG) layer, which conceptually is a simplified attention layer. Compared with attention, the MCG layer maintains global attention ability while keeping the computation linear in the number of *tokens*.

## 3.3 Interaction Modeling

We represent interactions using an energy model with parameters $\theta$, given input data $\mathcal{X}$. The probability is defined as:

$$p(\mathbf{s}|\mathcal{X}, \theta) = \frac{1}{\mathcal{Z}} \exp(-E(\mathbf{s}|\mathcal{X}, \theta)). \tag{1}$$

Specifically, a pairwise graphical model is used to compute energy as:

$$E(\mathbf{s}|\mathcal{X}, \theta) = \sum_i E_{traj}(\mathbf{s}_i|\mathcal{X}, \theta) + \sum_{(i,j)\in\mathcal{G}} E_{pair}(\mathbf{s}_i, \mathbf{s}_j|\mathcal{X}, \theta),$$

where unary potential is defined using predicted probability $q(\mathbf{s}_i|\mathcal{X}, \theta)$ from backbone as

$$E_{traj}(\mathbf{s}_i|\mathcal{X}, \theta)) = -\log(q(\mathbf{s}_i|\mathcal{X}, \theta)).$$

Different from previous approaches, we utilize a learned pairwise potential $E_{pair}$ with a dynamic underlying graph $\mathcal{G}$. For each agent pair $(i, j)$, we first project the predicted trajectories $\mathbf{s}_i$ into agent-j-centered coordinate system, denoted as $\mathbf{s}_{i@j}$, and vice-versa for $\mathbf{s}_{j@i}$. Note both $\mathbf{s}_{i@j}$ and $\mathbf{s}_{j@i}$ have $K$ trajectories for $i^{th}$ and $j^{th}$ agent respectively. Thus, the pairwise potential, as a $K \times K$ matrix, can be simply computed as:

$$E_{pair}(\mathbf{s}_i, \mathbf{s}_j|\mathcal{X}, \theta) = MLP([MLP([\mathbf{s}_{j@i}, \mathbf{s}_i]), MLP([\mathbf{s}_{i@j}, \mathbf{s}_j])]),$$

where $[]$ represents concatenation. The two inner MLPs share the weights. In our experiments, each MLP consists of two layers, the first with 128 and the second with 64 feature dimensions.

Intuitively, a pairwise potential provides the compatibility of the trajectory combinations between its two agents. For example, if the $a^{th}$ predicted trajectory of agent $i$ is colliding with the $b^{th}$ predicted trajectory of agent $j$, then $E_{pair}(\mathbf{s}_i, \mathbf{s}_j)[a, b]$ should have a much higher value, specifying that those two trajectories should not be selected at the same time to form one joint future prediction.

Although it is clear that not necessarily all agents in the graph $\mathcal{G}$ interact with each other, we do not have ground truth labels for the interactivity. While [35] proposed to use reinforcement learning to estimate the interactive graph, our core idea is to use the predicted marginal trajectory candidates. In particular, for each pair of agents $(i, j)$, if the most likely predicted trajectories $\mathbf{s}_i^a, \mathbf{s}_j^b$ are close with each other (i.e. their center locations are within the distance of their length and width), we connect these two agents in the interactive graph, *i.e.* $(i, j) \in \mathcal{G}$. In this way, the graph structure can be adapted for the specific scenarios in a dynamic fashion, thus we denote it as *dynamic graph* approach. In practice, we found this works especially well when the predicted trajectories are produced by a well trained backbone model. In one special case related to AV planning, where the task is to predict the AV's future trajectories and the relevant futures for the other agents, we utilize a star-shaped graph that is centered on the AV with connections to all other agents.

Given the full definition of the graphical model, we apply standard sum-product message passing to obtain the approximated joint probability as defined in Eq. 1 and max-product message passing to obtain the best joint trajectories for all agents. In practice, we instantiate 3 iterations of message passing in our models' compute graphs; we saw little benefit beyond this, which is a similar observation found in prior work with GNN interaction encoding [4].

**Conditional Setting:** Generating future trajectories explicitly conditioned on specific agents (usually AV) action is important for planning in practice. This has been explored by a few prior works [42, 46, 19, 24, 40], however, their focus is mostly on modeling the future of a single agent pair, instead of the full multi-agent setup. One advantage of leveraging a graphical model in our

| | Method | Overlap($\downarrow$) | minSADE($\downarrow$) | minSFDE($\downarrow$) | SMissRate($\downarrow$) | mAP($\uparrow$) |
|---|---|---|---|---|---|---|
| | LSTM baseline [20] | - | 1.91 | 5.03 | 0.78 | 0.05 |
| | HeatIRm4 [47] | - | 1.42 | 3.26 | 0.72 | 0.08 |
| | SceneTransformer(J) [19] | - | 0.98 | 2.19 | 0.49 | 0.12 |
| Test | M2I [40] | - | 1.35 | 2.83 | 0.55 | 0.12 |
| | DenseTNT [48] | - | 1.14 | 2.49 | 0.54 | 0.16 |
| | MultiPath++[32] | - | 1.00 | 2.33 | 0.54 | 0.17 |
| | *JFP [AV-Star]* | - | **0.88** | **1.99** | **0.42** | **0.21** |
| | SceneTransformer(M) [19] | 0.091 | 1.12 | 2.60 | 0.54 | 0.09 |
| Val | SceneTransformer(J) [19] | 0.046 | 0.97 | 2.17 | 0.49 | 0.12 |
| | MultiPath++[32] | 0.064 | 1.00 | 2.33 | 0.54 | 0.18 |
| | *JFP [AV-Star]* | **0.030** | **0.87** | **1.96** | **0.42** | **0.20** |

Table 1: WOMD Interactive Split: We report scene-level joint metrics numbers averaged for all object types over t = 3, 5, 8 seconds. Metrics minSADE, minSFDE, SMissRate, and mAp are from the benchmark [20]. In addition, we compute Overlap as defined in Sec. 4.1.

multi-agent interaction formulation is that it is easy to perform conditional inference, *i.e.* to generate joint future trajectories for other agents conditioned on a specific trajectory $j$ for some agent $i$. We can compute joints of all other agents using the same procedure as before but with modified unary potential for agent $i$; *i.e.* by setting the unary potential other than the conditioned trajectory to infinity ($E_{traj}(\mathbf{s}_i \neq j) = \infty$) to make agent $i$ always select trajectory $j$ during inference. Note this does not require separate training steps and is a natural extension of our inference procedure.

### 3.4 Loss Function and Optimization

We train our multi-agent motion forecasting model to minimize the negative joint log-likelihood of a given dataset $\mathcal{D}$:

$$\min_\theta \sum_{\mathbf{s}, \mathcal{X} \in \mathcal{D}} -\log p(\mathbf{s}|\mathcal{X}, \theta) \tag{2}$$

Instead of optimizing this objective function directly, previous methods [1, 9, 31] optimize the marginal probability for each agent, effectively decomposing the objective function into $\sum_{\mathcal{X} \in \mathcal{D}} \sum_i \log p(\mathbf{s}_i|\mathcal{X}, \theta)$. Then for each agent, a classification task with cross entropy loss is used for determining which one of the $K$ trajectories is favored, and a regression task with Huber loss is used to make the selected predicted trajectories closer to ground-truth.

In our work, while we utilize the same regression loss $\mathcal{L}_{reg}$ as in [32], we differ from previous methods on the classification loss, which is optimizing directly Eq. 2. The gradients of Eq. 2 w.r.t. all trainable weights can be computed with chain rule where first we derive that the gradients w.r.t. the unary and pairwise terms (*i.e.* $E_{traj}$, $E_{pair}$) are [2]:

$$\mathcal{L}_{E_{traj}} = \text{cross\_entropy}(\mu_i + \text{stop\_gradient}(\hat{\mu}_i - \mu_i))$$
$$\mathcal{L}_{E_{pair}} = \text{cross\_entropy}(\upsilon_{i,j} + \text{stop\_gradient}(\hat{\upsilon}_{i,j} - \upsilon_{i,j}))$$

Note to simplify the notation, we denote $\mu_i$ as the unary softmax logits and $\upsilon_{ij}$ as the pairwise softmax logits, *i.e.* $\mu_i = \log(q(\mathbf{s}_i|\mathcal{X}, \theta)) = -E_{traj}(\mathbf{s}_i|\mathcal{X}, \theta))$ and $\upsilon_{ij} = -E_{pair}(\mathbf{s}_i, \mathbf{s}_j|\mathcal{X}, \theta)$, and $\hat{\mu}_i, \hat{\upsilon}_{i,j}$ are corresponding marginal logits from message passing. Secondly, since $E_{traj}$ and $E_{pair}$ are directly output from the backbone network, we can easily backpropagate their gradients to all trainable weights. Thus, the optimization target of Eq. 2 can be replaced by the following loss function: $\mathcal{L} = \mathcal{L}_{reg} + \mathcal{L}_{E_{traj}} + \mathcal{L}_{E_{pair}}$.

## 4 Experiments

As we are proposing models for multi-agent motion forecasting in self-driving scenarios, it is essential to use large and comprehensive datasets with appropriate metrics. We first show experimental

---

[2] For detailed derivation, please refer to supplementary material.

|  | Overlap | | Marginal | | | | Pairwise | | |
| --- | --- | --- | --- | --- | --- | --- | --- | --- | --- |
|  | All | AV | minADE | minFDE | MissRate | mAP | minADE | minFDE | MissRate |
| SceneTransformer(M) [19] | 1.60 | 0.12 | 0.35 | 0.81 | 0.14 | 0.18 | 0.49 | 1.27 | 0.37 |
| SceneTransformer(J) [19] | 0.83 | 0.04 | 0.61 | 1.58 | 0.28 | 0.24 | 0.56 | 1.49 | 0.40 |
| MultiPath++ [32] | 0.99 | 0.05 | **0.30** | **0.74** | **0.03** | 0.47 | 0.37 | 1.00 | 0.11 |
| *JFP [AV-Star]* | 0.99 | **0.02** | **0.30** | **0.74** | **0.03** | 0.49 | **0.36** | **0.94** | **0.10** |
| *JFP [Dynamic]* | **0.79** | **0.02** | 0.31 | 0.76 | 0.04 | **0.49** | **0.36** | 0.96 | 0.11 |

Table 2: WOMD Extended: All-agents setting using standard track data.

results on Waymo Open Motion Dataset (WOMD) [20] for the interactive split, where we set the new state-of-the-art compared with the existing public methods. Next, we experiment with a more demanding "all-agent" setting using WOMD data, denoted as *WOMD Extended* described in detail below. Our method achieves significant improvements on joint metrics and is on-par or better on single-agent metrics. We also do an ablation study to investigate the choice of underlying interaction graph to provide more insights into our model. To further validate the effectiveness of our model, we experiment on a much larger (1000x) internal dataset and show improvements on all joint metrics, as well as demonstrate our model's applicability to the conditional prediction setting.

## 4.1 WOMD Interactive

This dataset provides 104K different scenarios, including 7.64 million unique tracks. Each data sample is a 9-seconds segment with 10Hz, where the first 1 second is used as inputs and the later 8 seconds are considered for prediction. High-definition 3D maps including traffic lights were provided as inputs. While the *standard track* focuses on single agent prediction, the *interactive track* is designed for evaluating joint prediction for one pair of agents in each scenario. Note that the selection process for the agent pair is designed to pick interacting agents [20].

For this dataset, we utilize a star-graph centered at a random agent for training, and star-graph centered at one of the target agents for inference (i.e. effectively only one edge since there are only two agents for evaluation). The prediction head of the backbone network consists of 5 ensemble heads that each provides 64 trajectories. NMS (Non-Max Suppression) is applied for each agent independently to generate the final 6 trajectories. We refer interested readers to [32] for more details. A star-graph is applied on top of the 6 output trajectories. Our model is trained on 16 TPU-v3 cores with batch size of 8*16 for 350K iterations ($\approx$ 20 hours). We use ADAMW with a learning rate of $4e^{-4}$ and decay to half only once at 200K iterations.

Table. 1 shows the results on both the validation and test sets. Numbers are shown for the official metrics averaged over object types and time steps. Our model clearly outperforms other public methods across all metrics.

**Overlap Metric** [3] We measure the consistency of predicted joint trajectories using the overlap metric. It is computed among the most likely trajectories for all target agents (two for each scene in this case). If the trajectories of a pair of agents overlap at any timestep, it is counted as one overlap. The final number is the sum of all overlaps in one scene, averaged over the dataset. The overlap metrics numbers in Tab. 1 show that our model is able to generate significantly more consistent predictions than the baselines.

## 4.2 WOMD Extended

While the WOMD *interactive track* provides good insights about two interactive agents, we further experiment with an all-agent setting where the number of target agents can be up to 40. The same data from WOMD is used for training and the validation data is obtained from the *standard track*, but includes all agents in the scene as target agents for joint modeling (up to 40 agents). This setup

---

[3]Note that the WOMD official metrics tool has an *overlap* entry, however that's only computing overlap between prediction and ground-truth but not among predicted trajectories from different agents. Thus it is not ideal for evaluating consistency among predictions.

| GraphType | Overlap | | Marginal | | | | Pairwise | | |
|---|---|---|---|---|---|---|---|---|---|
| | All | AV | minADE | minFDE | MissRate | mAP | minADE | minFDE | MissRate |
| None | 1.00 | 0.05 | 0.31 | 0.76 | 0.04 | 0.47 | 0.38 | 1.01 | 0.12 |
| Random-Star | 0.97 | 0.04 | 0.30 | 0.75 | 0.03 | 0.49 | 0.36 | 0.95 | 0.10 |
| AV-Star | 0.99 | **0.02** | **0.30** | **0.74** | **0.03** | **0.49** | **0.36** | **0.94** | **0.10** |
| Dynamic | **0.79** | **0.02** | 0.31 | 0.76 | 0.04 | **0.49** | **0.36** | 0.96 | 0.11 |
| Fully-Connected | 0.92 | 0.03 | 0.31 | 0.75 | **0.03** | 0.48 | 0.37 | 0.98 | 0.11 |

Table 3: WOMD ablation on using different interaction graphs.

| | Pairwise | | | | | Conditional Marginal | | | | |
|---|---|---|---|---|---|---|---|---|---|---|
| | Overlap | minADE | minFDE | MissRate | mAP | Overlap | minADE | minFDE | MissRate | mAP |
| MP++* [32] | 0.03 | 1.27 | 3.12 | 0.33 | 0.29 | 0.05 | 1.14 | 3.45 | 0.34 | 0.48 |
| CBP [24] | 0.03 | 1.23 | 3.02 | 0.32 | 0.30 | 0.02 | 1.05 | 3.24 | 0.33 | 0.51 |
| *JFP [AV-Star]* | **0.01** | **1.19** | **2.91** | **0.31** | **0.31** | **0.01** | **1.04** | **3.04** | **0.31** | **0.57** |

Table 4: Internal Dataset: JFP with AV-Star interaction graph shows consistent improvements over both joint metrics and conditional metrics, compared with competitive baselines including single-agent state-of-the-art Multipath++ [32] and a model designed for conditional setting CBP [24].

is closer to the full AV driving setting, and can provide extra insights for evaluating joint prediction models. In this experiment, we do not use ensembles for fast training speed.

**Metrics:** We use the aforementioned *overlap* metric, with *All* for overlap among all targets and *AV* for overlap involving the AV. However, overlap alone is insufficient, since a model can simply predict stationary futures for all agents to get 0 overlap. Thus standard single-agent marginal metrics are included as well, i.e. minADE, minFDE, MissRate, and mAP [20]. To better evaluate the joint model, we also introduce pairwise metrics, which naturally generalize the marginal metrics. For each pair of agents, we ask the model to provide a top-6 set of joint predictions, and compute the minSADE, minSFDE, sMissRate, and mAP. We only consider those pairs, whose agents have ever appeared within 10 meters to each other (according to ground-truth). This filtering can better differentiate interactive cases, since pairwise metrics for agents far away from each other highly correlate with marginal metrics.

Tab. 2 shows our models outperform MultiPath++[32] and SceneTransformer[19] by a large margin for overlap metrics while maintaining on-par or better marginal and pairwise metrics. Note our model with dynamic graph improves significantly over AV-Star on the *overlap* metrics, since it better captures the interactions among all agents.

**Ablation on Interaction Graph:** The performance of our model with different underlying interaction graphs are shown in Tab. 3, where *Random-Star* uses a star-graph centered at a random agent, *Fully-Connected* uses a fully-connected graph. We can see that the dynamic model significantly improves upon the baselines on the overlap metrics, without regressing the rest. While the *Fully-Connected* model has connections among all agents, it could be too complex and less accurate for message passing especially with dozens of agents, resulting in worse performance compared to the dynamic model.

**Qualitative Results:** In Fig. 2, we provide a few representative scenarios to demonstrate the effectiveness of our approach. Our model is able to output consistent future trajectories even for very crowded scenes.

## 4.3 Internal Dataset

Next, we conduct experiments on our internal dataset, which is about 1,000 larger than WOMD, to show how the methods generalize on industry-scale datasets. With this experiment, we are focusing on the joint prediction performance relative to the AV, which is most relevant for robust and safe planning. Thus we measure 1) pairwise metrics involving AV and 2) AV-conditional metrics. In this setup, we experiment with star-graphs compared against baseline models since they naturally fit to AV-centric applications.

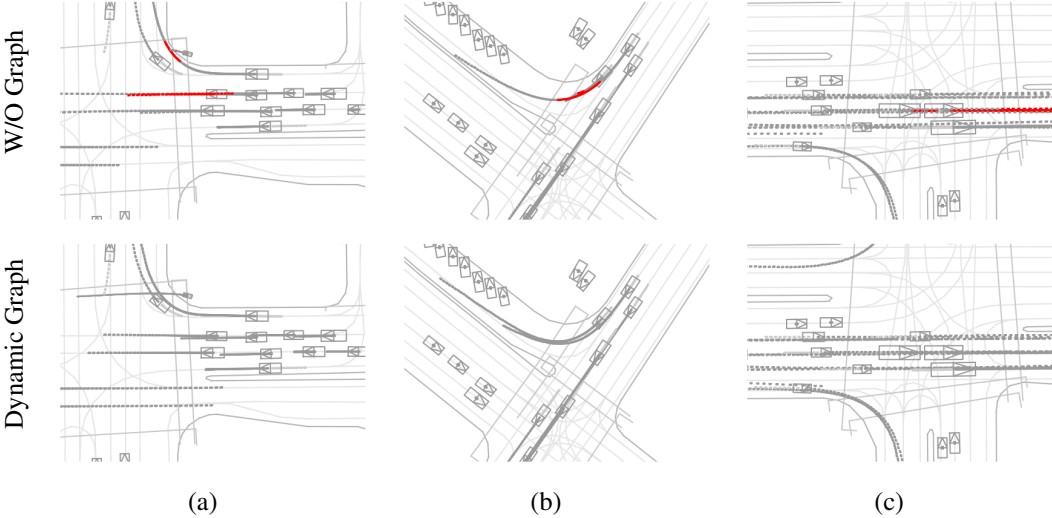

Figure 2: Visualization of most likely future predictions (8 seconds) for all agents on crowded scenes from Waymo Open Motion Dataset. We visualize each agent's predicted future trajectories as dark gray waypoints and its history trajectories as light gray waypoints. Trajectory overlaps among agents are marked with *red* for all corresponding future steps. Compared with the baseline model (top row), our approach (bottom row) can output consistent joint prediction without trajectory overlap on scenarios such as (a) at intersection, (b) for turning, and (c) with heavy traffic stack.

The dataset is prepared with the same setup as WOMD and our model is trained with the same hyperparameters as in Sec. 4.1 except 32 TPU-v3 cores are used with learning rate $1e^{-4}$. The MultiPath++ backbone is upgraded to handle the industrial-level dataset. The capacity of the encoders and the predictors are increased. The conditional model, CBP in Tab. 4, is our own implementation of [24] using the same backbone. Joint probability for CBP is calculated by $p(AV, other\ agent) = p(AV)p(other\ agent|AV)$. For the conditional metrics, we select the best AV trajectories (i.e. the prediction with the smallest minADE w.r.t. ground-truth) and compute metrics on all other the agents conditioned on these trajectories. Note for JFP, it can be computed as running conditional MRF inference to generate most likely trajectories for non-AV agents, while no extra change is needed for the MultiPath++ baseline since agents are independently predicted.

As shown in Tab. 4, our model consistently outperforms the baseline Multipath++ and CBP across all metrics by a large margin, even for conditional metrics. This shows our approach consistently has benefits on industrial-level dataset and models as well.

## 5 Limitation

The performance of our model could be bounded by how accurate the underlying interaction graph is, especially in sophisticated scenarios. While we proposed various graph types for different cases using human domain knowledge, it is still interesting to see if we can learn the interaction graph from data directly. However, this is non-trivial since there is no ground-truth for the graph. On the other hand, our approach does not do regression after message passing steps, which might limit the performance. We leave it to future work for the exploration.

## 6 Conclusion

We have proposed a deep structured model that can perform joint future predictions of multiple agents in a scene. Our model is based on pairwise MRF and can be trained end-to-end by optimizing directly the joint log-likelihood. The underlying interactive graph is dynamically generated from unary trajectories. Importantly, we show that our model can learn to avoid trajectory overlaps without corresponding domain knowledge. Experiments on public and larger scale internal datasets show significant improvement on joint motion forecasting metrics.

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
