# OpenReview forum: "JFP: Joint Future Prediction with Interactive Multi-Agent Modeling for Autonomous Driving"
_robot-learning.org/CoRL/2022/Conference — CoRL 2022 Poster_

### Official Review · Reviewer_2W5p · 2022-07-16

**Originality:** Good
**Technical Quality:** Good
**Clarity Of Presentation:** Good
**Impact:** 3

**Recommendation:**

Weak Reject: I recommend rejecting the paper, but will not argue for my recommendation if the majority of other reviewers have a different opinion.

**Summary:**

This paper proposed a deep structured model for multi-agent interactive trajectory prediction. It built upon typical multi-agent trajectory prediction model which predicts the marginal distributions of agents independently, and used an energy-based graphical model to predict the joint distribution of agents' trajectories. By adding the joint prediction layer, the proposed method outperformed the backbone MultiPath++ model and achieved the state-of-the-art performance on the interactive split of Waymo Open Motion Dataset.

**Issues:**

Please try to address the two concerns I mentioned before.

**Quality Of The Limitations Section:**

Additional details required

**Reviewer Expertise:**

5: The reviewer is absolutely certain that the evaluation is correct and very familiar with the relevant literature

**Robotics Focus:**

Highly relevant to robotics but no hardware experiments

**Strengths And Weaknesses:**

Strength: The proposed method is a sound and effective way to model the joint distribution of agents' trajectories. It can be easily plugged into any backbone models to improve the prediction performance in interactive scenarios.

Weaknesses:
1. While it makes sense to use the proposed energy-based graphical model to model the joint distribution of interacting agents, it is unclear if it is necessary and better than some simple heuristics. For example, what if $E_{pair}(\mathbf{s}_i,\mathbf{s}_j|\mathcal{X}, \theta)$ is simply set as zero if the predicted trajectory pairs do not overlap, and $+\infty$ if the predicted trajectory pairs overlap. The operation to verify trajectory overlapping is already inherited in the construction procedure of the dynamic graph, so it should a straightforward alternative of the proposed method. By doing so, it is equivalent to rule out all the combinations of trajectory candidates with collision and renormalize the joint distribution. I suspect it could work equally well as learning the energy function, because overlap rate seems to be the only metric that can be substantially improved by adding the graph structure, and the improvement is most effective when the model is informed which trajectory pair candidates overlap via the dynamic graph. I suggest the author to include this ablation study in the rebuttal or any other convincing evidences to show that the proposed method learns more than removing colliding candidates.

2. It is not clearly explained how to approximate the joint probability and sample from the joint distribution. While the authors briefly explain it in line 152-154, I am particularly curious about how the final predicted trajectories are selected from the modeled joint distribution. My current understanding is that $K$ trajectory candidates are drawn from the marginal distribution of each agent, the joint distribution is defined over $K^N$ combinations of candidates for $N$ agents. While the energy-based graphical representation and the heuristic dynamic graph reduce the computational complexity in computing $E(\mathbf{s}|\mathcal{X}, \theta))$, the probabilities of all the $K^N$ combinations need to be computed to find the ones with the highest probabilities. The dimension $K^N$ grows exponentially with the number of agents. The authors mentioned the number of agents could be up to 40 in their experiments. It will be great if the authors can provide more details on how the predicted trajectories are selected in their experiments.

**Summary Of Recommendation:**

The paper is generally well-written. The proposed method is sensible and can effectively improve prediction performance in interactive scenarios. However, it is still unclear to me whether the additional modules are indeed necessary and can be practically used for real-time trajectory prediction. At this stage, I am inclined to reject the paper since it is difficult to precisely evaluate the value of the paper with these unclear issues, but I am happy to change my score if the authors can provide convincing evidences in the rebuttal.

---

### Official Review · Reviewer_jdv8 · 2022-07-28

**Originality:** Good
**Technical Quality:** Good
**Clarity Of Presentation:** Good
**Impact:** 4

**Recommendation:**

Weak Accept: I recommend accepting the paper, but will not argue for my recommendation if the majority of other reviewers have a different opinion.

**Summary:**

This paper proposes a graph based decoder to do joint motion prediction by optimizing the pair-wise consistency by energy loss function. The performance of the proposed method is impressive and the 1000x larger experiments are interesting.

**Issues:**

- Unclear technical details: in Line 152-156, how the sum-product and max-product are performed, with what as input and output? How the iterations are done? Some experiments about the number of iterations would be appreciated.

- Missed reference:  [1, 2] are both joint motion prediction methods. Some comparison with them would be appreciated; in Line 134, you mentioned projection between different agents in multi-agent system, which is similar to [3, 4, 5]

- More datasets: as mentioned in the Weakness part, if experiments could be conducted on more public datasets, the work would have much larger impact and be easy for the community to follow. For example, I notice that [1,2] results on the INTERACTION challenge.

- More details: The 1000x internal dataset is impressive.  In Line 260-261, you mentioned that Multipath++ and the predictor has been upgraded to handle the industrial-level dataset. I am wondering under such large dataset, how the model scale. Some details such as hidden-dimension, number of layers, and number of iterations would be appreciated.

[1] Latent Variable Sequential Set Transformers For Joint Multi-Agent Motion Prediction. ICLR 2022

[2] THOMAS: Trajectory Heatmap Output with learned Multi-Agent Sampling. ICLR 2022

[3] Multi-Agent Trajectory Prediction by Combining Egocentric and Allocentric Views. CoRL 2021

[4] Roto-translated Local Coordinate Frames For Interacting Dynamical Systems. NeurIPS 2021

[5] MultiPath++: Efficient Information Fusion and Trajectory Aggregation for Behavior Prediction. ICRA 2022



**Quality Of The Limitations Section:**

Limitations are addressed clearly

**Reviewer Expertise:**

5: The reviewer is absolutely certain that the evaluation is correct and very familiar with the relevant literature

**Robotics Focus:**

Highly relevant to robotics but no hardware experiments

**Strengths And Weaknesses:**

Strengths:
- The proposed energy-based optimization goal for pair-wise consistency on the joint motion predicon task is novel.
- The performance on the larger scale dataset is impressive.
- The 1000x larger (compared to Waymo Open Motion) dataset is interesting

Weaknesses:
- The only open test protocol is Waymo Interaction Prediction, which only predicts for 2 agents jointly and might not be enough to fully examine the joint prediction performance in the real world setting. The Waymo Extended and internal dataset are both difficult for the community to follow. More experiments on public dataset/leaderboard would significantly improve the impacts of the work, for example, as mentioned in the paper, the INTERACTION challenge.
- Some of technical details are unclear. Please see Issues section for details.

**Summary Of Recommendation:**

In summary, the proposed method is intuitive and effective with impressive performance, which is why I give it accept. I appreciate the information about its 1000x larger internal dataset. However, it seems that it is not accessible to the research community.  Thus, I think some open source protocal code for Waymo Extended and experiments on other public datasets would make the work more influencial and easy to follow by the community. I would further raise my score if some of the two aforementioned efforts could be done or promised to be done.

---

### Official Review · Reviewer_CkUV · 2022-07-30

**Originality:** Very Good
**Technical Quality:** Very Good
**Clarity Of Presentation:** Excellent
**Impact:** 3

**Recommendation:**

Weak Accept: I recommend accepting the paper, but will not argue for my recommendation if the majority of other reviewers have a different opinion.

**Summary:**

The paper proposed a pairwise MRF on top of MultiPath++ to have consistent trajectory predictions that any pair of trajectories are encouraged not to overlap each other. It uses various types of graphs that determine the connection between trajectories in terms of energy and provides an ablation study. It achieves state-of-the-art performance in WOMD interactive and provides additional experiment results on additional datasets (WOMD extended, internal dataset).

[1] B. Varadarajan, A. Hefny, A. Srivastava, K. S. Refaat, N. Nayakanti, A. Cornman, K. Chen, B. Douillard, C. P. Lam, D. Anguelov, et al. Multipath++: Efficient information fusion and trajectory aggregation for behavior prediction. arXiv preprint arXiv:2111.14973, 2021.

**Issues:**

I have a few questions and comments.

* The main idea of the paper resembles the paper "Multimodal Trajectory Prediction via Topological Invariance for Navigation at Uncontrolled Intersections" [2] from CoRL 2020 as it modeled a joint probability of trajectories and considered a pair-wise relationship for consistent trajectory predictions. Adding descriptions that provide differences between the proposed method and the paper would be helpful for understanding the contribution. "Implicit Latent Variable Model for Scene-Consistent Motion Forecasting" [3] from ECCV 2020 also employed scene-level prediction with a joint probability model for trajectory prediction.
* For the dynamic graph construction, the connection is determined as described in line 145. I wonder if the affinity matrix or E_{pair} is initialized as an NxN matrix (like a fully-connected graph) and the connection determines non-zero values in the matrix. Also, "their center locations are within the distance of their length and width" sounds a little confusing. Is the trajectory s_{i} already predicted from MultiPath++? While defining the pairwise energy of s_{i} and s_{j}, it requires realized trajectories in its energy term. Is this determined by the ground truth trajectories? I am also curious how the distance metric is defined between trajectories. Even some trajectories that are not close enough may affect each other. For instance, if a vehicle in front of the AV is running fast, it may affect the AV's behavior of left turn.
* It seems that L_{reg} is not coming from Eq. 2 and is not defined in the paper. Is this for MultiPath++ model? Could you specify the term? Then how is this different from updating E_{traj}? Providing a clear distinction from MultiPath++ may help readers understand the paper.
* (optional) Visualization of pairwise energy from examples may help readers to see the effect of the term.
* (optional) What is the computation overhead of the graph approach? Though this is not the usual case to provide the speed of inference or training in this domain, providing some comparison of computation or speed of training or inference may help readers to understand the trade-off.
* (optional) Effective or the average number of connections may be another interesting point for readers. Different graphs were compared and the dynamic graph was effective since it reduces the complexity to attend to meaningful connections. Comparing effective connecting numbers of the dynamic graph w.r.t the distance threshold can help understand the trade-off in the number of connections vs performance. Also comparing the optimal numbers between different datasets can provide an indirect measure of the level of interactions for each dataset.


[2] Roh, J., Mavrogiannis, C., Madan, R., Fox, D., & Srinivasa, S. S. (2020). Multimodal trajectory prediction via topological invariance for navigation at uncontrolled intersections. arXiv preprint arXiv:2011.03894.
[3] Casas, S., Gulino, C., Suo, S., Luo, K., Liao, R., & Urtasun, R. (2020, August). Implicit latent variable model for scene-consistent motion forecasting. In European Conference on Computer Vision (pp. 624-641). Springer, Cham.

**Quality Of The Limitations Section:**

Limitations are not well addressed

**Reviewer Expertise:**

4: The reviewer is confident but not absolutely certain that the evaluation is correct

**Robotics Focus:**

Highly relevant to robotics but no hardware experiments

**Strengths And Weaknesses:**

Strengths
* Simple but an effective extension on MultiPath++ [1] for better modeling of interaction with a joint probability modeling.
* Show the improvement on various interactive datasets including a public benchmark.

Weaknesses
* Some details are missing: details of the model and hyperparameters (alpha and beta) for the reproduction of the numbers.
* Hard to distinguish what was coming from MultiPath++ and what was added as a contribution.

**Summary Of Recommendation:**

Though I had confused about the details of the method, the high-level idea and contribution to predicting trajectories of high interactions are clear and sound. While MultiPath++ provides a strong backbone for marginal trajectory prediction, the method provides a novel way to involve pairwise interactions in choosing the final trajectories with the pairwise MRF formulation and a joint probability modeling. It seems the proposed method is effective in predictions for interactive cases but similar to its backbone in marginal predictions since it uses MultiPath++ results as its initial candidates. As the details of the architecture (for MultiPath++) are not clear, the contribution of the paper might be limited to highly interactive cases; therefore I recommend weak acceptance.

---

### Official Review · Reviewer_DT7Z · 2022-08-02

**Originality:** Good
**Technical Quality:** Very Good
**Clarity Of Presentation:** Fair
**Impact:** 4

**Recommendation:**

Weak Reject: I recommend rejecting the paper, but will not argue for my recommendation if the majority of other reviewers have a different opinion.

**Summary:**

The paper introduces a new trajectory prediction method. It works as a layer on top of an existing forecasting method and creates a graph of agent-agent interaction that is used to improve prediction accuracy.
I overall like the paper and I verified that the authors submitted a new high score to the leaderboard but I don't fully understand the method.

**Issues:**

The method clearly works. Otherwise, it wouldn't be on the WOMD leaderboard.
But the writing and reproducibility need to be improved for CoRL standards.

**Quality Of The Limitations Section:**

Additional details required

**Reviewer Expertise:**

2: The reviewer is willing to defend the evaluation, but it is quite likely that the reviewer did not understand central parts of the paper

**Robotics Focus:**

Relevant but unlikely to deploy to hardware in near future

**Strengths And Weaknesses:**


### Strengths

- **S.1** The writing is mostly great and the structure is good. (Except for the method section as noted below.)
- **S.2** The authors do beat the high score on the WOMD Interaction Challenge. This is a hard task, so I think this is great evidence that this method works.
- **S.3** The accompanying video is good at visualizing the differences between a baseline and this method


### Weaknesses

- **W.1** I'm going to have to veto the title. "SMART" is already taken as acronym in the motion forecasting literature and it's from 2 years ago: https://arxiv.org/pdf/2007.13078.pdf Creating another method with the "SMART" acronym will only cause confusion in the community. Plus for their title, the acronym works significantly better and they don't have to pull the "R" from the middle of a word. ;)
- **W.2** Clarity. Even on the second read, the method isn't intuitively clear to me and I don't know how much of that is my fault. I don't hold this fully against the authors and I hope they can clarify this during the rebuttal. From what I understand, you predict pairwise interactions and then you use some message-passing magic to use these pairwise interaction probabilities to improve predictions. But how? (a) The pair-wise interaction predictor is supervised by distance, right? If 2 agents are within each other's threshold at timestep, then what's the supervisory signal for their "pairwise potential"? (b) And how does the message-passing work? How is that implemented, what data travels back and forth between agents? (c) How is either of that used to generate new/different trajectories? Are you fine-tuning the backbone with this somehow or are you modifying the predicted trajectories? (d) I really think it'd help to present a bit of code here, either as proof-of-concept Jupyter notebook or as pseudocode.
- **W.3** Results on internal dataset/reproducibility. We obviously can't reproduce the results from your internal dataset. We don't have any specifics for this dataset. We don't know how much work was put into the baselines. For all intents and purposes, you could have completely made up these numbers. I would just cut that part of the paper and replace that with implementation details.
- **W.4** Generality. If I understand correctly, this method is supposed to work with any kind of backend, right? Why don't you show that? SceneTransformer outperforms MultiPath++ on the WOMD Interactive dataset, so could you run an experiment with that as backend? If the argument is that code isn't available, then why not the similar AutoBots [29]? If your method does not work with other backends or if you can't show that, then I'd rephrase some sections of your paper to reflect that it's an extension of MultiPath++.

**Summary Of Recommendation:**

I'm happy to adjust my rating if the authors explain their method a bit more and respond to the other weaknesses noted above.

---

### Meta-Review · Area_Chair_JT2g · 2022-08-05

**Recommendation:** Accept (Poster)
**Confidence:** 4

**Metareview:**

This paper introduces a motion forecasting head that better models interactions between agents for motion forecasting. The approach shows significantly improved performance, particularly in settings with extensive interaction. Post rebuttal reviewers all agreed this paper warrants acceptance and commended the good results on a number of challenging real world datasets, and the addition of useful ablations/ additional clarity about the method. I think the paper is much improved as a result of reviewer feedback and recommend acceptance.

For the camera ready paper, I encourage the authors to add the additional ablations/ and experiments conducted on additional datasets to the supplementary material, and include methodology clarifications in the main paper.

I commend both authors and reviewers for their conduct in the review process, which I believe is a good example of a highly successful rebuttal and discussion phase.